# Opioids and Breast Cancer Recurrence: A Systematic Review

**DOI:** 10.3390/cancers13215499

**Published:** 2021-11-01

**Authors:** Merlino Lucia, Titi Luca, Del Prete Federica, Galli Cecilia, Mandosi Chiara, De Marchis Laura, Della Rocca Carlo, Piccioni Maria Grazia

**Affiliations:** 1Department of Maternal and Child Health and Urological Sciences, Sapienza University of Rome, Policlinico Umberto I, Viale del Policlinico 155, 00161 Rome, Italy; lucia.merlino@uniroma1.it (M.L.); cecilia.galli@uniroma1.it (G.C.); chiara.mandosi@uniroma1.it (M.C.); mariagrazia.piccioni@uniroma1.it (P.M.G.); 2Department of Anesthesiology, Critical Care and Pain, Section Obstetrical Care, Sapienza University of Rome, Policlinico Umberto I, 00161 Rome, Italy; luca.titi@hotmail.it; 3Department of Radiological, Oncological and Pathological Sciences, Sapienza University of Rome, Policlinico Umberto I, 00161 Rome, Italy; laura.demarchis@uniroma1.it; 4Department of Medical-Surgical Sciences and Biotechnologies, Sapienza University of Rome, Policlinico Umberto I, 00161 Rome, Italy; carlo.dellarocca@uniroma1.it

**Keywords:** opioids, anesthesia, breast cancer, breast cancer surgery, breast cancer recurrence

## Abstract

**Simple Summary:**

Opioids are one of the therapeutic and palliative options for breast cancer, a tumor with a strong epidemiological impact. Studies have been extensively reported in the literature that show a connection between the administration of opioids and the recurrence of the disease, both during surgery and subsequently in the management of cancer pain. This argument, in consideration of the strong impact of this cancer, is of great interest. Therefore, we decided, through the study of the existing literature, to describe the state of the art on this topic, to outline the best therapeutic approach to be adopted in these delicate patients.

**Abstract:**

Breast cancer has the greatest epidemiological impact in women. Opioids represent the most prescribed analgesics, both in surgical time and in immediate postoperative period, as well as in chronic pain management as palliative care. We made a systematic review analyzing the literature’s evidence about the safety of opioids in breast cancer treatment, focusing our attention on the link between opioid administration and increased relapses. The research has been conducted using the PubMed database. Preclinical studies, retrospective and prospective clinical studies, review articles and original articles were analyzed. In the literature, there are several preclinical in vitro and in vivo studies, suggesting a possible linkage between opioids administration and progression of cancer disease. Nevertheless, these results are not confirmed by clinical studies. The most recent evidence reassures the safety of opioids during surgical time as analgesic associated with anesthetics drugs, during postoperative period for optimal cancer-related pain management and in chronic use. Currently, there is controversial evidence suggesting a possible impact of opioids on breast cancer progression, but to date, it remains an unresolved issue. Although there is no conclusive evidence, we hope to arouse interest in the scientific community to always ensure the best standards of care for these patients.

## 1. Introduction

Breast cancer has the greatest epidemiological impact in women, affecting 2.1 million women each year, and it is responsible for the major cancer-related deaths among women. In 2020, there were 2.3 million women diagnosed with breast cancer and 685,000 deaths globally, meaning approximately 15% of all female cancer deaths [1]. For the treatment of breast cancer, the crucial therapeutic moment is the surgery time, despite the several therapeutic options available to date. Paradoxically, the perioperative time is a very delicate moment because of the risk of metastasis development [2]. The perioperative period includes three moments: a pre-operative period corresponding to a few preoperative hours, an intraoperative period and a postoperative period consisting of several days after surgical treatment [3]. Three surgical factors favor the process of initiation and progression of residual tumoral cells: firstly, surgery itself, because the surgical manipulation may inadvertently scatter residual cancer cells into the bloodstream and lymphatic circulation [3,4] and determines a state of immunosuppression by depressing cell-mediated immunity (CMI). In addition, surgery shifts the balance towards angiogenesis, increasing vascular endothelial growth factor (VEGF) and other several growth factors and reducing antiangiogenic factors such as angiostatin and endostatin [5]. Secondly, general anesthetics contribute to impair immune functions for macrophages, dendritic cells, T lymphocytes and NK cells [6]. Thirdly, there is evidence in the literature that the use of opioid analgesics could inhibit immune function, angiogenesis, and tumor growth [7].

The combination of surgical stress and anesthesia causes a suppression of immune defense mechanisms in the perioperative period [8,9], which occurs within a few hours during surgery and lasts for several days [3,10].

Opioids represent the most prescribed analgesics in cancer patients, both in surgical time and in the immediate postoperative period, but they also have a central role in chronic pain management and at end-of-life as palliative care.

The use of opioid drugs is necessary when cancer pain is moderate or intense and the choice of the appropriate opioid depends on the intensity of the pain, clinical status of the patient, nociceptive component (somatic, visceral, or neuropathic), presence of metastases and type of tumor [11,12].

The reference opioid to treating cancer pain is morphine in all its formulations; it is still today the first choice to control moderate–severe cancer pain because it has effective and good pain control, absence of a ceiling effect and a low toxicity profile.

In mild pain, it is recommended to start treatment with weak opioids (codeine and tramadol) while when the intensity of pain increases, the clinical choice is on stronger opioids: fentanyl, buprenorphine, oxycodone, hydromorphone, buprenorphine [13]. In the literature, there is controversial evidence suggesting a possible impact of opioids on cancer progression. To date, whether opioids could favor cancer recurrence or metastasis is still an unresolved issue. Moreover, most studies investigated the hypothesis that regional anesthesia/analgesia in cancer surgery could reduce the cancer recurrence rate compared with general anesthesia alone. Indeed, regional anesthesia prevents the neuroendocrine stress response by interrupting both afferent neural transmissions to the central nervous system and descending efferent activation of the sympathetic nervous system (SNS). Thereby, regional anesthesia decreases endogenous opioids’ secretion and the requirement of opioid administration, reducing the opioid-induced immune suppression [14]. There are no sufficient data to support or refute the use of regional anesthesia technique, such as paravertebral block (PVB), for the reduction of cancer recurrence or improvement in cancer-related survival. However, PVB seems to be associated with lower levels of inflammation and a better immune response in comparison with general anesthesia and opioid-based analgesia [15].

The aim of this systematic review is to trace the main evidence regarding the risks of opioids in cancer patients’ treatment to identify the most recent accredited theory, paying particular attention to the risk of recurrence of the disease. In fact, additional aims are to investigate the influence of opioids on the immune system cells activity, on the process of angiogenesis and on cell proliferation/death to better understand the role of opioids in the tumorigenesis process.

## 2. Materials and Methods

This systematic review was conducted according to the PRISMA guidelines with adherence to all items except those related to the conduction of a meta-analysis (#15, #16, #21 and #23) [16].

### 2.1. Eligibility Criteria

The study question was structured according to PICO (Population: patients undergoing breast cancer surgery; Intervention: primary cancer surgery with or without use of opioids; Comparison: perioperative and/or chronic use of opioids vs. not; Outcome: overall mortality and overall free survival). All human and animal in vitro and in vivo studies meeting the PICO criteria were eligible for inclusion. Language was limited to English. No limitations were set on study design or publication year.

### 2.2. Study Collection and Selection

A systematic search has been conducted in the PubMed database and was updated until August 2021. The keywords used for the search were: “opioids” or “opioids anesthesia”, “opioids analgesia”, “general anesthesia” and “regional anesthesia” and “breast cancer”, “breast cancer recurrence” and “breast cancer outcomes”. Based on these terms, preclinical studies, retrospective and prospective clinical studies, review articles and original articles of interest were analyzed regarding perioperative and chronic use of opioids in breast cancer patients. Two investigators screened abstracts and titles of all articles independently. After the primary screening, articles (*n* = 1729) were full text screened and discussed between two investigators. A total of 185 articles were assessed for eligibility, of which 161 were excluded due to language limitations, study setting, study design and study outcomes. We excluded articles that include study populations affected by non-breast cancers, cancer patients not undergoing surgery and patients not treated with opioids. There were no limitations on the patient’s age, ethnicity, staging and cancer histotype. The inclusion criteria were breast cancer patients, patients undergoing surgery and patients perioperative or chronically treated with opioids. We also included in vitro studies which investigated the interaction between opioids treatments and immune cells activity, angiogenesis, and opioids’ action on cell proliferation/death.

Finally, only 24 studies were included in our systematic review (Figure 1). Conflicts were evaluated in consensus with a third investigator.

## 3. Discussion

Our literature review identified 24 relevant articles, of which 16 studies investigated the relationship between the immune system and opioids (*n* = 10), opioids and angiogenesis (*n* = 2) and opioids and tumor growth (*n* = 4). The remaining 8 clinical studies investigated the association between breast cancer recurrence and opioids, with a total of 41,591 cases. In this research, we considered articles that investigated patients with breast cancer undergoing opioids use or not. Outcomes of interest were recurrence-free survival or overall survival in relationship to opioids use. Four articles demonstrated that the use of opioids is not associated with a significant change in RFS or/and OS. Four studies showed that opioids use reduced the risk of breast cancer recurrence.

### 3.1. Opioids and Immune System

A total of 10 selected studies, of which 7 preclinical studies (Table 1) and 3 randomized clinical studies (Table 2), investigated the linkage between opioids and the immune system.

The immune system has a crucial role in the process of cancerization especially for cancer cells’ ability to evade the immune defenses [27]. The first line of defense against cancer progression and metastasizing is cell-mediated immunity (CMI), which includes cytotoxic T-cells (CTLs), dendritic cells, macrophages, and natural killer (NK) cells. Indeed, they have the feature to recognize non-self-cells, including tumor cells. The ability of NK cells in preventing cancer genesis and eradicating tumor cells confers them a central role in lots of studies about cancer recurrence [17,18]. There is a complex interaction between opioids and the immune system, and this relationship is controversial. Several studies showed that opioids interfere with immune cells activity; for instance, it was shown that fentanyl can inhibit macrophages and NK cells [19]. Shavit et al. showed that morphine can produce a suppression of natural killer cell cytotoxicity (NKCC) reversible with naloxone, in a dose-related manner [24]. Moreover, Beilin et al. showed that a large dose of fentanyl or sufentanil can suppress NKCC too [20,21]. This phenomenon is also detectable in patients receiving large doses of fentanyl anesthesia, in which NKCC results are significantly suppressed, although this does not happen in patients receiving small doses of fentanyl anesthesia [28].

Furthermore, opioids have been related to neutrophil migration [29] and reduced cytokine production [30]. It has also been shown that immune-competent cells express opioid receptors and go through apoptosis when treated with opioids [3]. Therefore, opioids with suppression of the immune system and with suppression of NK cells may predispose patients to a major risk of metastasizing, especially after cancer surgery, which is itself a source of immunosuppression [31]. The perioperative period represents a stress which promotes the inflammatory response and suppression of cell-mediated immunity [2,25]; the neuroendocrine and immunomodulatory responses underlying this process have been well decoded [32]. A stimulation of the hypothalamic-pituitary-adrenal (HPA) axis and SNS by surgery, anesthetics and analgesics agents occurs during the perioperative period. This neuroendocrine activation leads to an increase of catecholamines, prostaglandins, glucocorticoids, and opioids. All this causes a depression of NK cells and CTLs, in association with a decrease in interleukin-12 (IL-12), tumor necrosis factor-α (TNF-α), and interferon-γ (IFN-γ), which shift the balance T-helper1(Th1)/T-helper 2 (Th2) in favor of a Th2 anti-CMI environment [26]. Acting together, factors such as surgical stress, endocrine changes, and anesthetics used during the perioperative period can cooperate to create a particular favorable milieu in which cancer cells can evade our immune system and can disseminate in the bloodstream, increasing the risk of recurrence.

Several studies have investigated if using regional anesthesia/analgesia in cancer surgery may produce a reduced rate of recurrence or metastasis, because of their minor impact on the immune system. They examined opioid sparing anesthesia (such as regional anesthesia) and its impact on recurrence rate in comparison to general anesthesia alone. Sultan et al., in a controlled randomized study analyzing blood samples from patients undergoing paravertebral analgesia (PVA) and sevoflurane/opioid anesthesia, showed that PVA can attenuate the cytokine changes caused by the surgical stress by producing an increase in serum levels of IL-12 and IFN-γ and a decrease of IL-6 and IL-10. Thus, higher levels of IFN-γ/IL-12, important cytokines for the CMI-response, are maintained during the perioperative period in favor of a Th1 environment [22].

Several studies were conducted on samples from the ongoing randomized clinical trial (RCT) by Sessler et al. [23], which was analyzing the impact of breast cancer recurrence in patients undergoing general anesthesia with volatile anesthetic (sevoflurane) and opioids and patients undergoing regional anesthesia/analgesia with paravertebral block and propofol. Desmond et al., studying breast cancer tissues from both groups of patients, demonstrated that tissues from the propofol-paravertebral anesthesia group had increased tissue infiltration of NK and T-helper cells compared with the sevoflurane/opioid group [33]. Furthermore, it was observed that serum from patients undergoing PVA from the same RCT produced apoptosis to a greater degree in ER-negative breast cancer cells compared with sevoflurane/opioid anesthesia group [34]. Nevertheless, Sessler et al. concluded that perioperative use of opioids is not associated with an increased risk of breast cancer recurrence [23]. Furthermore, analgesia may interfere with immune system and cancer progression both directly, intervening in cellular mechanisms such as apoptosis, or indirectly by acting on neuroendocrine and sympathetic systems: indeed, it has been demonstrated that there was a reduction of pulmonary metastases after surgery in animal models treated with morphine-based analgesia [35,36]. These results indicate that the choice of anesthetic technique seems to have profound implications on the cytokine background and immune microenvironment in breast cancer surgery, and that this may consequently influence cancer recurrence and metastasis [3].

### 3.2. Opioids and Angiogenesis

Two selected preclinical studies investigated the association between opioids and angiogenesis (Table 3).

Angiogenesis is a strategy activated by the tumor to receive nourishment and to grow, as well as to evade the host’s defenses [38]. Therefore, it is strictly associated with the etiology and pathogenesis of tumor progression and metastasis [37,39,40,41]. Cancer cells autonomously secrete VEGF in an autocrine and paracrine way to create a new capillary and lymphatic network [42].

It was proven that morphine, at clinically relevant doses, promotes angiogenesis by increasing microvascular endothelial proliferation, survival and cell cycle progression, both in in vitro and in vivo assays. This translates in an increased vascularization of human breast tumor xenografts in mouse models, as demonstrated by Gupta et al. [38]. The administration of methylnaltrexone (MNTX), a peripheral mu-opioid receptor (MOR) antagonist, can reverse this effect [43]. In addition to this mechanism, surgical stress provokes an increase in catecholamines levels, which in turn determines an overexpression of angiogenic factors, such as VEGF, and matrix metalloproteinase 2 and 9 (MMP 2/9) and proinflammatory cytokines, such as IL-6 and IL-8 [44,45]. Th2 cytokines upregulate the expression of arginase-1 by myeloid-derived suppressor cells (MDSCs) in lymphoid tissue, leading to an arginine deficient environment, which impairs lymphocytes’ function through the synergistic action of prostaglandin E2 (PGE2) [46]. Moreover tumor-derived PGE2 induces MDSCs and tumor-associated macrophages (TAM, M2 phenotype), creating an immunosuppressive and proinflammatory environment that contributes to tumor angiogenesis [47,48]. Studies have shown that morphine can also promote invasion and migration of cancer cells by upregulating matrix metalloproteinases (MPs) in breast cancer [49].

### 3.3. Opioids and Tumor Growth

A total of four selected preclinical and clinical studies examined the relationship between opioids and tumor growth (Table 4). Opioids bind to specific receptors distinguished in mu, kappa, delta, and opioid-like receptors [50]. Mu opioid receptors are expressed pre- and post-synaptically and intervene in the nociceptive pathway. There are a variety of different mu-opioid receptors (MOR), binding different opioid ligands, which in turn could determine different responses [51,52]. It has been demonstrated that MOR is expressed in various cancers and increased cytoplasmic levels may be associated with a higher risk of lymph node metastases [53].

An in vitro study by Weingaertner et al. demonstrated a significant reduction in Heregulin-induced cellular growth and increase in cellular apoptosis determined by a chronic treatment with morphine in a human epidermal growth factor receptor 2 (HER2)-positive human breast cancer cell line [54].

In the Michigan Cancer Foundation-7 (MCF-7) breast cancer cell line, Methylnaltrexone showed anti-proliferative activity. Moreover, at its therapeutic concentrations for opioid-induced constipation, methylnaltrexone may enhance the tumoricidal activity of 5-Fluorouracil (5-FU). This effect is most likely to be attributed to a complementary action of methylnaltrexone, which reduces the expression of cyclin A, a S-phase of cell cycle regulatory protein, which induces mitosis. However, it is not clear if the growth reduction mediated by m-receptor antagonists is due to a competition with estrogen receptors or via other mechanisms [56]. Indeed, peripheral MOR antagonists, such as naltrexone and methylnaltrexone, are structurally like 17-b-oestradiol (E2). Therefore, they can compete with the estrogens for binding to their specific receptors, which are expressed in 80% of human breast cancer, inhibiting their promoting action of cancer cells’ survival and angiogenesis [57]. However, there is also evidence demonstrating the opposite phenomenon. In fact, Maneckjee et al. showed that the treatment of MCF-7 breasts tumors, which express different types of opioid receptors, with opioid ligands leads to a tumor growth reduction. Whereas the treatment with naloxone, an m-opioid antagonist, reverses this effect [58].

In triple negative breast cancer, lacking HER2, estrogen or progesterone receptors, the opioids growth factors (OGF) and its receptor pathway is present and regulates cell proliferation both physiologically and pathologically. OGF is a constitutively expressed peptide which inhibits DNA synthesis upregulating cyclin-dependent inhibitory kinases and its receptor intervenes in the proliferation of triple-negative breast cancer cells. This pathway could represent a possible therapeutic strategy to investigate. In fact, an intermittent low-dose naltrexone treatment in a cell culture model of a highly aggressive, invasive and poorly differentiated triple-negative breast cancer has been shown to promote the production of endogenous OGF, which results in inhibition of DNA synthesis and a 35% reduction in cell numbers [59].

According to some evidence, opioids’ action on cell proliferation/death seems to depend on the concentration and duration of treatment. In fact, low concentrations or single doses seem to promote tumor growth, while high concentrations or chronic treatment result in a growth-inhibitory effect [55,60]. Other in vitro studies showed that morphine and other opioids at elevated concentrations, not clinically possible, could present a negative influence on tumor growth by promoting apoptosis, avoiding angiogenesis, and inhibiting matrix metalloproteinase expression [61,62]. In transgenic mice with breast cancer, it has been demonstrated that morphine is not associated with tumorigenesis but with the progression of an existing tumor [63]. Nevertheless, in clinical practice, the relationship between cancer recurrence and opioid use remains controversial.

### 3.4. Chronic Use of Opioids

Several studies investigated the impact of opioids in the perioperative period in cancer recurrence, but less is known about the long-term effects of opioid use in cancer survivors. This field may require clarification because of the importance of this drug’s prescription in chronic pain management in patients with a comorbidity.

Cronin-Fenton et al., in an observational study including 34,188 women with stage 1 to 3 incident breast cancer, showed that there was no evidence between post diagnosis opioids prescription and cancer recurrence, and these results do not differ significantly according to opioid strength, cumulative dose or chronic long-term exposure or in analyses stratified by tumor estrogen receptor status, menopausal status and type of surgery. Patients underwent semiannual follow-up examinations for the first 5 years after diagnosis and annual examinations for the next 5 years [64]. Moreover, the same study showed a reduction of the recurrence rate in patients undergoing strongly immunosuppressive opioids, but these findings were probably due to an enhanced rate of mortality in this cohort, which thereby may have masked the recurrence cases. Indeed, Ekholm et al. in this cohort showed that the risk of all-cause mortality was 1.72 times higher among long-term opioid users than among individuals without chronic pain [65]. Thus, the authors attribute this result to channeling bias [66]; moreover, in this kind of patient, pain management can hide symptoms of cancer recurrence.

A recent US-based observational study including a cohort of 4216 women, observed for a median follow-up of 6.3 years, showed how common chronic use of opioids is, namely around 3.2% during the years following diagnosis and treatment of breast cancer, and includes 10% of breast cancer survivors. They also demonstrated that there is no significant increase of breast cancer recurrence in breast cancer survivors making chronic use of opioids and reassure them about the safety of these drugs [67].

In conclusion, there is no evidence about avoiding opioid use in breast cancer survival, and all the studies conducted on chronic users encourage the safety of these drugs which are so common and so helpful in pain management.

### 3.5. Clinical Data: Retrospective and Prospective Studies

A total of eight clinical studies investigated the linkage between opioids administration and breast cancer recurrence (Table 5).

One of the most relevant studies about the association between cancer recurrence or metastases and anesthetic technique is the retrospective analyses by Exadaktylos. The study showed a significant reduction in the risk of recurrence or metastasis four-fold during a 2.5- to 4-year follow-up period in patients treated both with paravertebral block and general anesthesia versus patients who received general anesthesia alone followed by morphine analgesia (94% vs. 82% at 24 months and 94% vs. 77% at 36 months after surgery, respectively) [68].

Cata et al., instead, found no differences regarding outcomes (recurrence free survival and overall survival) in patients treated with PVB versus patients treated with opioid-based analgesia [69].

Forget et al., in their retrospective analysis, investigated the effect of the administration of different intraoperative analgesics (sufentanil, ketamine and ketorolac) on cancer recurrence. They found, in contrast with previous studies, that sufentanil had no deleterious effect on cancer recurrence [70].

These findings are supported by the retrospective study of Lee et al., who did not find any deleterious effect of perioperative opioids on the recurrence of cancer between two groups of patients receiving propofol-based total intravenous anesthesia (TIVA) or sevoflurane-based anesthesia. The first group received higher concentrations of opioids. No differences in overall survival were detected in the two groups, whereas a lower rate of cancer recurrence was reported in the propofol group [71].

Kim et al. found that use of tramadol as a rescue analgesic after breast cancer surgery was associated with a decreased risk of postoperative disease recurrence rate (7.3% in the tramadol group vs. 9.4% in the control group) and lower mortality (4.7% vs. 8.1%, *p* = 0.001) [72].

To date, there have been only two prospective studies in the literature investigating the association between the use of opioids anesthetics/analgesics and recurrence of breast cancer. A Danish population-based cohort study by Cronine-Fenton et al. did not find an association between postdiagnosis opioid administration and breast cancer recurrence, regardless of opioid type, strength, chronicity of use or cumulative dose [64]. Similarly, Sessler et al. have done the most recent multicenter randomized trial, in which 2132 women were enrolled, of whom 1043 were assigned to regional anesthesia-analgesia and 1065 were allocated to general anesthesia. The authors investigated the impact of breast cancer recurrence in patients undergoing general anesthesia with volatile anesthetic sevoflurane and opioids compared with regional anesthesia/analgesia using paravertebral block and propofol. Median follow-up was 36 (IQR 24–49) months. The authors reported an equal breast cancer recurrence rate equal to 10% in both groups, concluding that the perioperative use of opioids is not associated with an increased risk of second breast cancer events [22]. Lastly, in a very recent study, Montagna et al. performed a retrospective analysis of 1143 triple negative breast cancer (TNBC) cases undergoing opioids anesthesia and found a protective effect of intraoperative opioids on recurrence-free survival in TNBC. In fact, a higher intraoperative opioid dose was associated with favorable recurrence-free survival, hazard ratio 0.93 (95% confidence interval 0.88–0.99) per 10 oral morphine milligram equivalents increase (*p* = 0.028), but was not significantly associated with overall survival, hazard ratio 0.96 (95% confidence interval 0.89–1.02) per 10 morphine milligram equivalents increase (*p* = 0.2). Therefore, opioid receptor expression analysis showed an upregulation of antitumor receptors and a downregulation or no expression of protumor receptors [73].

## 4. Conclusions

In summary, opioids remain not only the most frequent analgesic treatment in cancer pain management, but also a fundamental anesthetic choice in surgical time. The beneficial results deriving from the opioids use in cancer patients consist of an improvement in quality of life, a reduction in neuroendocrine stress responses and in emotional stress, which in turn provides a better compliance with cancer treatments, all resulting in a positive influence on survival.

Given the heterogeneity of the evidence available to date, it is not clear if opioids may or not influence the cancer outcomes. In the literature, there are several preclinical, in vitro and in vivo studies suggesting a possible linkage between opioids administration and progression of cancer disease or metastasizing. Nevertheless, these results are not confirmed by clinical studies, and the insufficient evidence does not support the avoidance of such important drugs in the management of breast cancer disease. Indeed, according to the results of the most recent randomized trial, opioids should continue to be used both during surgery as analgesics associated with anesthetic drugs, and in the postoperative period for optimal cancer-related pain management. However, well-designed prospective studies are necessary to clarify the association between opioid use and overall survival across breast cancer.

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
