# Peer review of "Opioids and Breast Cancer Recurrence: A Systematic Review"

_cancers, 2021, doi:10.3390/cancers13215499_

Round 1

Reviewer 1 Report

Merlino Lucia &Titi Luca et al has reviewed the association between Opioids and breast cancer recurrence. The authors have retrieved 1729 articles and analyzed them systematically. The review is precise and well written with lots of relevant information. However, there are few minor suggestions to strengthen the article.

Comments:

  • In introduction section, the authors must consider to briefly explain the level, frequency and preference of opioids use in cancer treatments.
  • The authors screened 1729 records, however 1544 were excluded just based on title and abstract. Finally, only 24 studies were considered for analysis. Exclusion and Inclusion criteria must be elaborated more in detail for the easy understanding of the readers.
  • Being a review article, results section looks very minimal, and part of this information could have been included in methodology section. Indeed, the authors must consider merging results and discussion section.

Author Response

Thank you for appreciating our work and for the valuable advice. The anesthetist first author of the work has expanded the section dedicated to the use of opioids in cancer. We have refined the methodology section by better explaining the inclusion and exclusion criteria. Finally, as recommended, we merged the results section with the discussion section. We hope that the work we have done can satisfy all requests and that it can be even more appreciated.

Reviewer 2 Report

Comments:

The introduction describes well the information known so far.

  • The citation of the reference with similar content and design  is missing (Pérez-González O, Cuéllar-Guzmán LF, Soliz J, et al.Impact of Regional Anesthesia on Recurrence, Metastasis, and Immune Response in Breast Cancer Surgery: A Systematic Review of the Literature. Regional Anesthesia & Pain Medicine 2017;42:751-756.)

The aim of the study is defined only as studying the safety of opioids in cancer treatment, which is not a description of what has been studied -link to the immune system, angiogenesis, tumor growth and disease outcome.

  • Usually drug safety refers to the frequency of adverse drug effects (i.e., physical or laboratory toxicity that could possibly be related to the drug) that are treatment emergent. This is not the case in this study, where it is looking for a link between opioids and the outcome of the disease. Please clarify!

The materials and methods are appropriate and understandable. Scheme 1 captures the investigated literature well.

The results are summarized in detail in tables that are relevant and help the reader understand the content.

  • I suggest dividing Table 1 into preclinical trials and randomized clinical trials.

The results are adequately commented on in the discussion and are also appropriately placed in context with similar findings. Citation is appropriate.

Author Response

Thank you for appreciating our work and for your valuable advice. We have cited the recommended work in the introduction. We have tried to better clarify the object of study by modifying the aim of the work. We have divided table 1 into two tables in order to make the work clearer and more immediate.We hope that the improvements made can be adequate and that the work can thus be satisfactory.